nanotechnology/materials science

multiple polymerization, seed emulsion polymerization, poly(acrylic acid), magnetic nanospheres

**Author for correspondence:**
Kaimin Chen
e-mail: kmchen@sues.edu.cn

This article has been edited by the Royal Society of Chemistry, including the commissioning, peer review process and editorial aspects up to the point of acceptance.

# Synthesis of poly(acrylic acid) coated magnetic nanospheres via a multiple polymerization route

Kai Li[1], Kaimin Chen[1], Qiaoling Wang[1], Ying Zhang[2] and Wenjun Gan[1]

[1]College of Chemistry and Chemical Engineering, Shanghai University of Engineering Science, Shanghai 201620, People's Republic of China
[2]State Key Laboratory of Chemical Engineering, School of Chemical Engineering, East China University of Science and Technology, Shanghai 200237, People's Republic of China

KC, 0000-0002-1238-7340

Magnetic nanospheres are versatile candidates for both fundamental and practical applications. Before they are applied in more complicated fields, their surface must be modified by several functionalities. However, the surface modification can be affected by the magnetic nanoparticles (MNP) embedded in the polymer matrix. Herein, the synthesis of poly(acrylic acid) coated magnetic nanospheres via a multiple polymerization route is described. During the synthesis process, seed emulsion polymerization was applied to redistribute the MNP in the polymer matrix, and the relationship between the structure of magnetic nanospheres and the thickness of the grafted poly(acrylic acid) layer was investigated. The development of size, morphology and magnetic properties of the nanospheres were characterized by transmission electron microscopy, dynamic light scattering, thermogravimetric analysis, X-ray diffraction and vibrating sample magnetometry. This work would pave the way to design and preparation of new structure of functional magnetic nanospheres with precise surface modification.

## 1. Introduction

Magnetic polymer nanospheres combine the special properties of inorganic magnetic nanoparticles (MNP) and functional polymeric materials, which have many potential applications [1,2] in the treatment of waste oil pollution in the ocean [3], drug delivery [4–6], separation and purification of biological materials [7], contrast agents [8], adsorption of heavy metal ions [9], and hyperthermia [10]. Among these applications, the magnetic

**Scheme 1.** Schematic illustration of the preparation of the poly(acrylic acid) coated magnetic nanospheres (PMNS).

polymer nanospheres are mainly used as a carrier to bind materials such as biomolecules and can be easily manipulated by an external magnetic field. For example, Oster *et al.* used poly(vinyl alcohol)-based magnetic beads to separate genomic DNA and purify sequence specific nucleic acid [11]. Nishio *et al.* synthesized magnetic nano-carriers with poly(glycidyl methacrylate) on the surface, which could immobilize many types of drugs and could be used in separation of proteins by affinity purification [12]. Therefore, the functional group modification on the surface of the magnetic nanospheres greatly enhances their performance in a variety of applications.

Researchers have reported various functional groups on magnetic polymer nanospheres, such as hydroxyl groups [13], amino groups [14–16], thiol groups [17], epoxy groups [18,19] and carboxyl groups. Among them, there are many interesting works on the study of carboxyl groups [3,20–22]. Carboxyl groups can be used as a carrier for catalysts [23,24], agent for adsorption of proteins [25,26], and template for the synthesis of metal nanoparticles [27,28]. Carboxyl groups also provide high stability in water and can be stored for a long time [22]. However, it was found that the surface modification by functional groups was greatly affected by the employed synthesis route in the preparation of magnetic nanospheres.

In previous studies, researchers have been working on the development of various polymerization methods, such as suspension polymerization [29,30], precipitation polymerization [31], emulsion polymerization [32], miniemulsion polymerization [33–37], soap-free emulsion polymerization [38], to control the morphology of magnetic polymer nanospheres. In these studies, miniemulsion polymerization is considered to be the most effective way to encapsulate inorganic nanoparticles [35]. In order to further improve the encapsulation of MNP, some researchers have adopted the method of multi-step polymerization. They either synthesized magnetic beads firstly and then immobilized the carboxyl-containing nanoparticles onto the magnetic bead surface by self-assembly [39,40] or used more complicated methods to improve the morphology of magnetic polymer nanospheres [19,41]. However, there is still a lack of a cost-effective way to balance the surface modification of nanospheres and MNP existing in the polymer matrix. Therefore, we provide an alternative way to prepare functional magnetic nanospheres.

In our work, a multiple polymerization route was used to prepare poly(acrylic acid) (PAA) coated magnetic nanospheres (PMNS). A strategy combining three polymerization methods including miniemulsion polymerization, seed emulsion polymerization and photoemulsion polymerization was used here (scheme 1). Especially, seed emulsion polymerization was applied to control the MNP distribution in polystyrene. At the same time, the effect of aggregation state of MNP in magnetic nanospheres on surface functionalization was also studied.

# 2. Material and methods

## 2.1. Chemicals

Iron acetylacetonate (Fe(acac)$_3$), oleic acid (OA), oleylamine, absolute ethanol, biphenyl ether, *n*-hexane, sodium dodecylsulfate (SDS), hexadecane (HD), and methacryloyl chloride (MC) were purchased from Adamas Reagent Co., Ltd. 2-Hydroxy-4′-hydroxyethoxy-2-methylpropiophenone (HMP) (Irgacure 2959) was purchased from SANN Chemical Technology Shanghai Co., Ltd. 1,2-Dodecanediol was bought from Tokyo Chemical Industry Co., Ltd. Potassium persulfate (KPS) was obtained from Shanghai Lingfeng Chemical Regent Co., Ltd. Styrene (St) and acrylic acid (AA) (from Shanghai Titan Scientific Co., Ltd) were distilled under reduced pressure and placed in a 4°C refrigerator for later use. The

photoinitiator 2-[$p$-(2-hydroxy-2-methylpropiophenone)]-ethylene-glycol-methacrylate (HMEM) was prepared according to previous work [20].

## 2.2. Preparation of magnetic nanoparticles

MNP were prepared by a pyrolysis process [42]: Fe(acac)$_3$ (1.42 g), biphenyl ether (40 ml), 1,2-dodecanediol (4.05 g), OA (3.39 g), and oleylamine (3.21 g) were mixed in a 250 ml three-necked flask. Under nitrogen atmosphere and magnetic stirring, the temperature was raised to 200°C and kept for 2 h. Then the temperature was raised to 260°C, refluxed for 2 h and cooled down to room temperature. An amount of 80 ml of absolute ethanol was added to the black mixture and centrifuged at 10 000 rpm for 20 min to obtain a black precipitate. To purify MNP, the black sediment was redispersed in 10 ml of $n$-hexane (0.1 ml of OA and 0.1 ml of oleylamine were added simultaneously) and the residue was removed by centrifugation at 6000 rpm for 10 min. Finally, the MNP were obtained through re-precipitation with ethanol, and were redispersed in $n$-hexane (20 wt%), kept at a low temperature (4°C) for later use.

## 2.3. Preparation of seed magnetic nanospheres

An amount of 0.01 g of SDS was completely dissolved in 10 g of deionized water. A mixture of 0.05 g of HD, 1 g of styrene and 0.05 g magnetic fluid (20 wt%) was made in a test tube, followed by ultrasonication with a frequency of 53 kHz and a power of 350 W for 15 min. The sonicated mixture was then placed in a 30 ml sample vial and mixed with SDS solution and then homogenized (Ultrasonic Cell Shredder from Ningbo Xinzhi Biotechnology Co., Ltd) for 10 min to obtain a pre-emulsion. An amount of 5 ml of KPS solution (0.002 g ml$^{-1}$) was mixed with the pre-emulsion and added to a 100 ml three-necked flask. The miniemulsion polymerization was carried out under nitrogen protection at 80°C with mechanical stirring of 300 rpm for 20 h. The emulsion was then subjected to dialysis treatment for further use. These magnetic nanospheres synthesized by miniemulsion polymerization are denoted as seed magnetic nanospheres (MNS1).

## 2.4. Preparation of magnetic nanospheres coated by photoinitiator

A mixture of SDS solution (0.002 g solid) and KPS solution (0.005 g solid) was added into a three-necked flask. A suitable amount of MNS1 emulsion was then added as a seed emulsion, and extra deionized water was added to make the total amount of water 20 g. The mixture was mechanically stirred at 300 rpm in a nitrogen atmosphere, and the temperature was raised to 80°C. Then, 1 g of styrene, loaded in a micro-injector (model TJ-1A/L0107-1A, Longer), was injected slowly with a rate of 1 ml h$^{-1}$ to the three-necked flask. Five hours after the addition of styrene, 0.5 g of photoinitiator (HMEM) in acetone (mass ratio of 1 : 5) was added to the system at a rate of 10 min ml$^{-1}$ in the dark. After continuous reaction for 12 h, a thin layer of photoinitiator was coated on the magnetic polystyrene core. The magnetic nanospheres synthesized by seed emulsion polymerization are denoted as magnetic nanospheres coated by photoinitiator (MNS2).

## 2.5. Synthesis of poly(acrylic acid) coated magnetic nanospheres

MNS2 and AA monomer ($m_{MNS2} : m_{AA} = 1 : 1$) were added to a homemade photoreactor and the mixture was diluted to 1.0 wt% by deionized water. The reaction was started under UV lamp irradiation and completed in 2.5 h with vigorous magnetic stirring.

## 2.6. Characterization

The apparent hydrodynamic size and size distribution of the samples were obtained by dynamic light scattering (DLS) by NICOMP 380 ZLS measured at a fixed scattering angle of 90°. X-ray diffraction (XRD) analysis was performed with a Bruker D8 Advance X-ray diffractometer with a scan speed of 5° min$^{-1}$ and an angular range of 10° to 90°. Thermogravimetric analysis (TGA) with a model Q600 SD was used to analyse the magnetite content at a heating rate of 10°C min$^{-1}$ in air atmosphere. The hysteresis loop of the magnetic samples was characterized by a vibrating sample magnetometer (VSM, model Squid-VSM) with a test temperature of 298 K and a magnetic range of $-20$ kOe to 20 kOe. High-resolution transmission electron microscope (TEM) images were obtained at 200 kV using a JEOL-2100F.

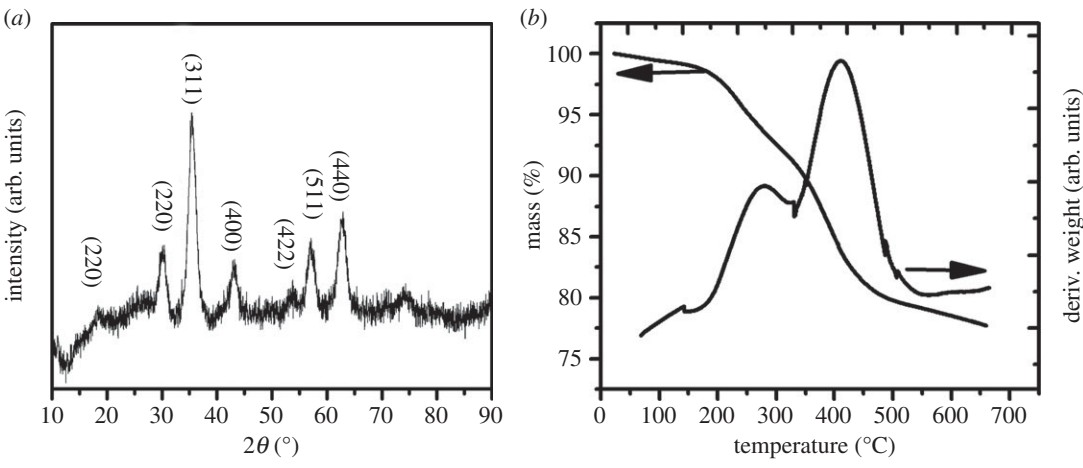

**Figure 1.** (*a*) XRD spectrum and (*b*) thermogravimetric analysis curve of MNP.

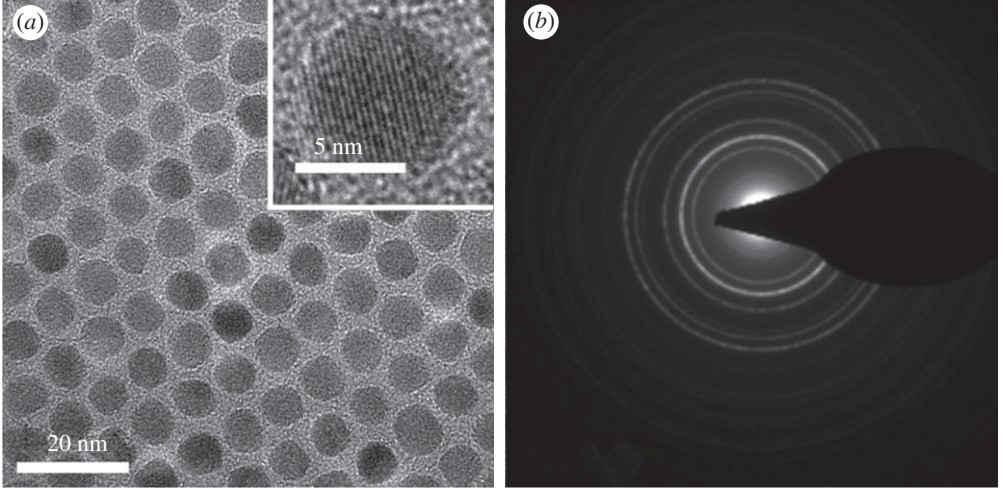

**Figure 2.** (*a*) TEM image and (*b*) selected area electron diffraction of MNP. Inset in (*a*) is a high-resolution image of a single MNP.

# 3. Results

## 3.1. Synthesis of magnetic nanospheres

### 3.1.1. Magnetic nanoparticles

In order to obtain MNP with uniform size and excellent magnetic properties, pyrolysis method was applied to synthesize oleic acid and oleylamine modified MNP. Excess oleic acid and oleylamine were used to obtain a highly stable magnetic fluid based on MNP. The structure of MNP was characterized by XRD as shown in figure 1*a*, and there are seven characteristic peaks at positions of 18.7° (220), 30.3° (220), 35.5° (311), 43.2° (400), 53.7° (422), 57.1° (511), 62.9° (440), which are consistent with the results reported in the literature [43,44]. The MNP size can be calculated by Scherrer formula based on XRD result and it is 8.9 nm [45]. TGA was employed to study the ligands (oleic acid and oleylamine) content on MNP surface and the result is shown in figure 1*b*. It was found that the total mass loss of oleic acid and oleylamine was 23% by weight, and there are two decomposition stages observed from the DTG curve in figure 1*b*. The corresponding decomposition temperatures are 257°C and 410°C, and the weight loss is 9% and 14%, respectively, indicating that there are two bonding ways of ligands on the surface of MNP. The weight loss at low temperature (110°C to 308°C) is caused by physical adsorption and the weight loss at high temperature (308°C to 530°C) is caused by chemical bonding [46,47].

TEM was used to observe the size and shape of the MNP as shown in figure 2. It showed that the diameter of the particles is about 9 nm, which is consistent with the calculated result from the XRD pattern in figure 1*a*. From the inset in figure 2*a*, a uniformly oriented lattice with explicit periphery

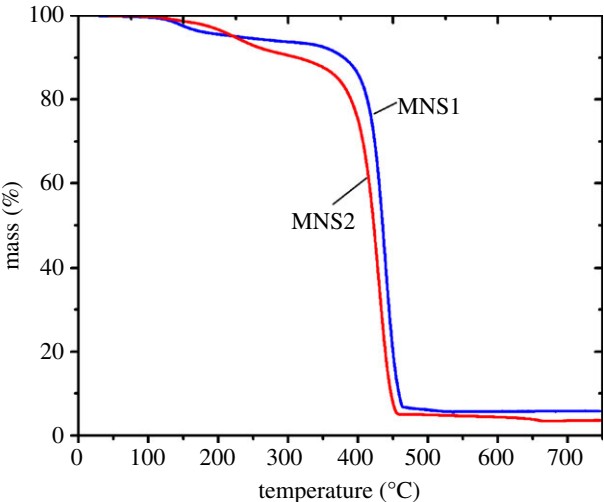

**Figure 3.** Thermogravimetric analysis curves of MNS1 and MNS2.

can be seen in a single MNP, indicating that each particle is a single crystal. The gap between each MNP is around 2 nm, which is ascribed to ligands (oleic acid and oleylamine) on MNP surface. Figure 2*b* shows the selected area electron diffraction mode obtained for 9 nm MNP, in line with the results of XRD.

### 3.1.2. Characterization of MNS1 and MNS2

Miniemulsion polymerization is known to be the best choice for encapsulating various types of nanoparticles, including MNP. But it was reported that MNP could move to the near surface of the nanosphere, which will influence further surface functionalization like immobilization of carboxyl groups [37]. Seed emulsion polymerization is known as an excellent tool to increase the seed size as well as redistribution of inorganic nanoparticles inside the seed [19]. It was reported that the particle morphology by seed emulsion polymerization was affected by the polymerization process [48]. There are two alternative processes, a swelling method and a semi-continuous feeding method. It is considered that monomer added by the swelling method initiates polymerization inside the seed, and the MNP would appear on the near surface or segregated on one side of the nanosphere due to the exclusion between different substances [41]. Compared to the swelling method, the semi-continuous feeding method is known to control the monomer feeding rate to reduce the matrix swelling to some extent and ensure the seed polymerization occurring on the seed surface [48]. Therefore, in order to reduce the adverse redistribution of MNP, the semi-continuous feeding method was chosen to add monomers during the seed emulsion polymerization process.

Figure 3 shows a TGA diagram of MNS1 and MNS2. The magnetite content is 5.9 wt% and 4.3 wt% for MNS1 and MNS2, respectively. Compared to MNS1, the decreased magnetite content of MNS2 is a result of increased content of polystyrene and photoinitiator HMEM. TEM is also used to characterize the change of size and morphology before and after seed emulsion polymerization. From the TEM images as shown in figure 4*a,c*, the size is *ca* 150 nm and 200 nm for MNS1 and MNS2, respectively, which is consistent with the data obtained from DLS measurements (figure 4*b,d*). More uniform nanospheres containing MNP could be seen in figure 4*c* and the polydispersity index of DLS also confirmed this finding ($0.17 \pm 0.02$ for MNS1 and $0.11 \pm 0.01$ for MNS2). Averagely, the hydrodynamic size increased by 50 nm from MNS1 to MNS2. Both the decrease in MNP content and the increase in the diameter of the nanospheres confirmed the successful seed emulsion polymerization of MNS2 from MNS1.

### 3.1.3. Hysteresis loops

Magnetic properties of MNP, MNS1 and MNS2 at 298 K were recorded and compared using a VSM as shown in figure 5. In figure 5*a*, the saturation magnetizations of MNP, MNS1 and MNS2 are 52.4 emu g$^{-1}$, 2.7 emu g$^{-1}$ and 0.96 emu g$^{-1}$, respectively. In figure 5*b*, the saturation magnetization value of MNP is 68.0 emu g$^{-1}$ (ligands were subtracted according to TGA results). The inset in the upper left corner shows a small residual magnetization (0.75 emu g$^{-1}$) and coercive force (30 Oe), at which MNP are considered to be superparamagnetic [49]. In figure 5*b*, the saturation magnetizations of

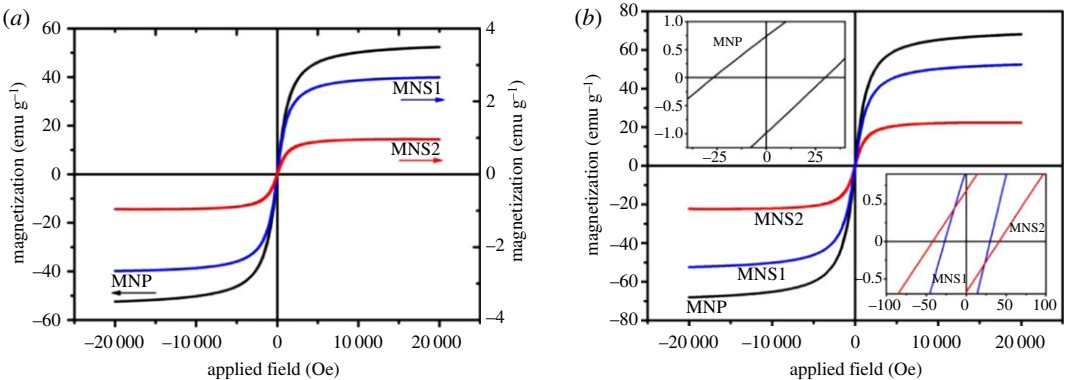

**Figure 4.** TEM images of (*a*) MNS1 and (*c*) MNS2, and hydrodynamic sizes of (*b*) MNS1 and (*d*) MNS2.

**Figure 5.** (*a*) Hysteresis loops and (*b*) hysteresis loops excluding the ligands and polymers of MNP, MNS1, and MNS2 at 298 K. The insets are enlarged plots.

MNS1 and MNS2 were reduced to 45.1 emu g$^{-1}$ and 22.3 emu g$^{-1}$ (ligands and polymer were subtracted according to TGA results), respectively. The residual magnetizations and coercive forces of MNS1 and MNS2 remain within the superparamagnetic range. Compared with MNP, the saturation magnetizations of MNS1 and MNS2 were reduced by 33.7% and 67.2%, respectively. After polymer encapsulation, the nanospheres still show superparamagnetism, but the saturation magnetization decreases as polymer content increases. The decrease in the saturation magnetization may be ascribed to the fact that the higher-powered ultrasonic operation causes partial oxidation of the magnetite nanoparticles to form non-magnetic iron oxide during the emulsifying process. In addition, the encapsulation of the polymer shell may also result in a loss of magnetization [50].

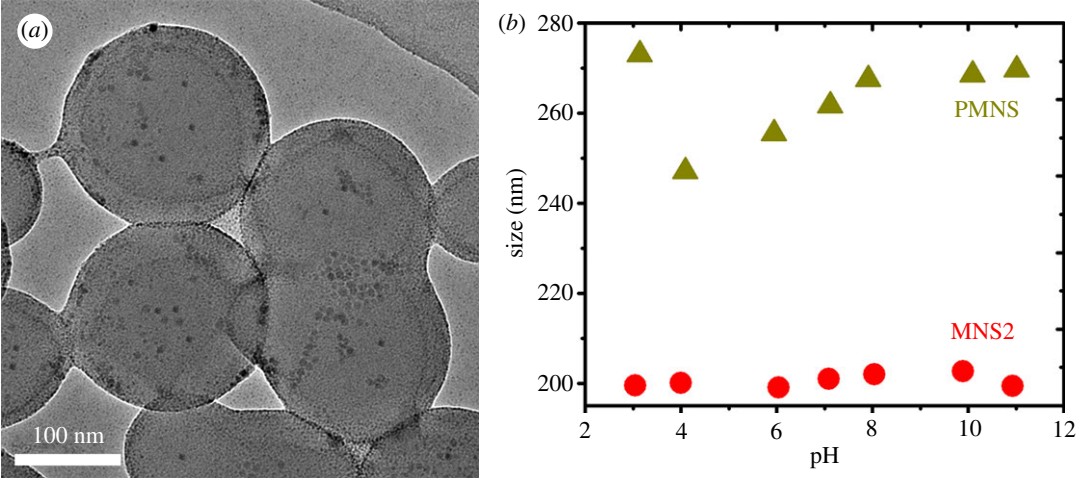

**Figure 6.** (*a*) TEM image of PMNS and (*b*) size of MNS2 and PMNS as a function of pH.

**Table 1.** Dosage and layer thickness in each step during the synthesis of PMNS.

| entry | styrene dosage ($m_{St}/m_{MNS1}$) | L1 (nm) | AA dosage ($m_{AA} : m_{MNS2}$) | L2 (nm) | L (nm) |
|---|---|---|---|---|---|
| 1 | 1.02 | 2.0 | 1 : 1 | 39.2 | 41.2 |
| 2 | 2.05 | 5.5 | 1 : 1 | 37.5 | 43.0 |
| 3 | 3.02 | 7.5 | 1 : 1 | 36.8 | 44.4 |
| 4 | 10.01 | 26.6 | 1 : 1 | 18.3 | 44.8 |

## 3.2. Synthesis of poly(acrylic acid) coated magnetic nanospheres

### 3.2.1. Poly(acrylic acid) grafting on MNS2

AA is chosen as a model functional monomer bearing negative carboxyl groups and was grafted on MNS2 surface by photoemulsion polymerization to obtain PMNS. Since the near surface of MNS2 is coated with a thin layer of photo-initiator HMEM, under the irradiation of ultraviolet light, AA is polymerized from the surface of MNS2 to form a brush-like PAA shell, which is reported in the literature as spherical PAA brushes [20]. In figure 6*a*, the size of diameter observed by TEM image changes little compared with that of MNS2 due to relatively low grafting density of PAA [51]. While, the successful PAA grafting can be characterized by DLS as shown in figure 6*b*. With the increase of pH value, the hydrodynamic size of MNS2 is almost a constant, because MNS2 has no pH sensitivity. As the pH increases, the thickness of the PAA layer increases, which is ascribed to deprotonation of carboxyl groups and electrostatic repulsion between negative charges. PMNS is unstable at low pH ($pH \leq 4$) because of insufficient space hindrance and reduced electrostatic repulsion. The MNS2 showed a large amount of precipitation in one month. While the PMNS stored in a PBS buffer was stable for up to 6 months without precipitation, which demonstrated that the PAA chains endow PMNS with good stability. Compared to literature [52], however, the PAA length is relatively short under the same conditions. Considering that MNP may affect the photoinitiator encapsulation and then further PAA grafting, the relationship between shell increase in the MNS2 and PAA grafting in the PMNS will therefore be investigated.

### 3.2.2. Effect of seed emulsion polymerization on layer thickness of poly(acrylic acid) coated magnetic nanospheres

To reduce the MNP effect on further surface functionality of MNS2, styrene dosage was gradually increased as shown in table 1. The different particle size was calculated from DLS data and the corresponding sizes are illustrated in scheme 2. It was found that the size of polystyrene layer (L1) increased with the increase of

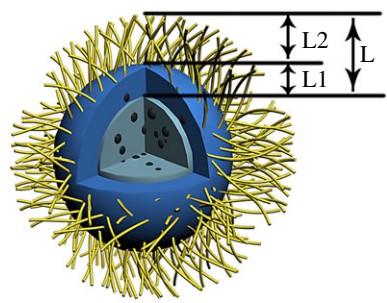

**Scheme 2.** Schematic illustration of different layers in the preparation process of PMNS. L1 means the increased size of polystyrene layer from MNS1. L2 means the increased size of PAA layer from MNS2. L means the combination of L1 and L2.

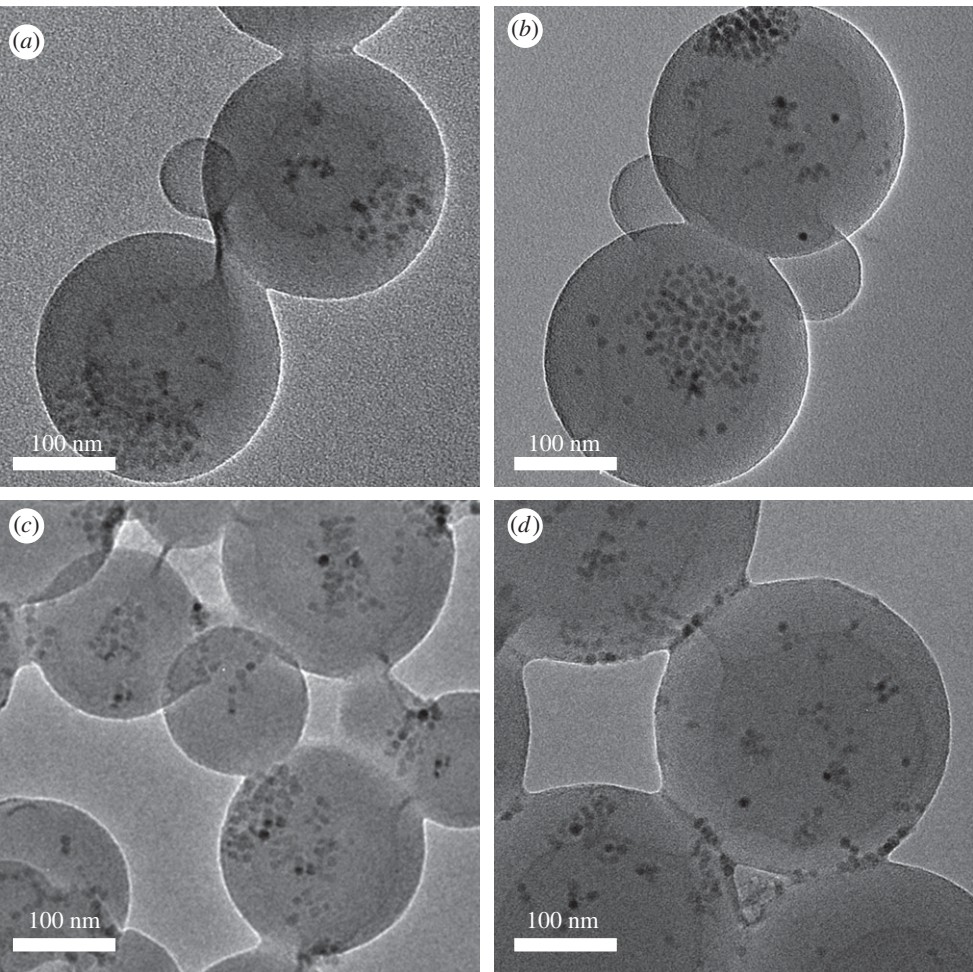

**Figure 7.** TEM images of (*a*) MNS1, (*b*) MNS2 of Entry 1, and (*c*) MNS1, (*d*) MNS2 of Entry 4.

styrene dosage, indicating more polymer was coated by seed emulsion polymerization mechanism. In the photoemulsion polymerization, the AA monomer dosage was kept as a constant ($m_{AA} : m_{MNS2} = 1 : 1$). It was surprising to find that the size of PAA layer (L2) decreased with the increase of styrene dosage in the seed emulsion polymerization. In order to figure out what happened during the multiple polymerization, TEM images of Entry 1 and Entry 4 were compared. For Entry 1 with a low styrene dosage, as shown in figure 7*a,b*, it was found that MNP appeared to be a cluster both in MNS1 and MNS2. But, for Entry 4 with a higher styrene dosage, MNP showed a more dispersed state inside MNS2 (figure 7*d*) compared to MNS1 (figure 7*c*). It indicates that excess styrene monomer may penetrate into the interior of MNS1 and initiate polymerization, resulting in a redistribution of MNP. Part of MNP would migrate to the near surface of MNS2, directly affecting the photoinitiator coating and further grafting efficiency of the PAA

on MNS2 surface. It is also found that the combined size ($L = 43 \pm 2$ nm) of L1 and L2 is approximately a constant for all entries in our experiments. Therefore, the total particle size is a balance between MNP distribution and PAA amount which was ascribed to the monomer penetration to nanospheres matrix when excess styrene monomer was added. Xu *et al.* used a seed emulsion polymerization method to increase the thickness of functional layer by approximately 10 nm [19]. Compared with the literature, our method improves the layer thickness to some extent.

# 4. Conclusion

PMNS containing MNP in the core and bearing PAA chains on the surface were prepared via a multiple polymerization route. MNP modified with oleic acid and oleylamine were synthesized by a pyrolysis route with a uniform size of 9 nm and ligand content of 23 wt%. The saturation magnetization of magnetite reached 68 emu g$^{-1}$ and it is superparamagnetic. Seed emulsion polymerization by semi-continuous feeding method was successfully carried out on MNS1 to obtain MNS2. Both MNS1 and MNS2 showed superparamagnetic property but with decreased saturation magnetization by 33.7% and 67.2%. The pH sensitive PAA was finally grafted to MNS2 via photoemulsion polymerization. There was a balance between polystyrene layer thickness in MNS2 and PAA layer thickness in PMNS. Hence, we provide a novel multiple polymerization route to synthesize PMNS which would be very useful in a variety of areas like magnetic separation, biomedical field, etc.

Data accessibility. All relevant data are within the paper and raw data are in the electronic supplementary material.
Authors' contributions. K.L. and K.C. established the experimental programme, conducted the trial work and drafted the manuscript together, and thus made an equal contribution to the study. Q.W. provided schematic drawing and preparation of experimental materials. Y.Z. proposed the experimental improvement programme. W.G. helped draft and revised the manuscript. All the authors gave their final approval of the version submitted for publication.
Competing interests. We declare we have no competing interests.
Funding. This research was funded by National Natural Science Foundation of China grant no. 21504052 and Shanghai Sailing Program grant no. 15YF1404800.
Acknowledgements. We are grateful to the National Natural Science Foundation of China and the Science and Technology Commission of Shanghai Municipality for providing sufficient financial support for our research.

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
