## [Reviewer comments · Royal Society Open Science]

Review History

RSOS-190141.R0 (Original submission)

Review form: Reviewer 1

Is the manuscript scientifically sound in its present form?

Yes

Are the interpretations and conclusions justified by the results?

Yes

Is the language acceptable?

Yes

Is it clear how to access all supporting data?

Yes

Do you have any ethical concerns with this paper?

No

Have you any concerns about statistical analyses in this paper?

No

Recommendation?

Accept with minor revision (please list in comments)

Comments to the Author(s)

In this manuscript, Chen, et al, reported a multiple polymerization route to synthesize poly (acrylic acid) coated magnetic nanospheres with controlled surface modification. The obtained magnetic nanospheres were fully characterized by XRD, TGA, TEM, DLS, and VSM. Besides, the relationship between the structure of magnetic nanospheres and the thickness of the grafted poly (acrylic acid) layer has been investigated. This work provided a new idea to control the structure and surface of magnetic polymer nanospheres. Therefore, I recommend that this work can be published in RSOS after addressing some specific issues as follows:

1. Line 13, Page 3: the description on Figure 1 is not correct. The peak positions for (422), (511), and (440) is wrong, there three peaks should be 53.7°, 57.1°, and 62.9°.
2. Page 4, line 32. Authors described the pH responsiveness of magnetic nanospheres, however, it's better to provide the hydrodynamic size in extreme pH values such as pH 3 and pH 11 in Figure 6.
3. The magnetic content of the nanospheres is low. Can the loading amounts of magnetite in polymer matrix be adjusted?
4. How the stability of the poly (acrylic acid) coated magnetic nanospheres in PBS?
5. In the caption of Figure 5, "loop" should be "loops"
6. Some related reported papers on the surface modification of magnetic composites (such as Chem. Commun. 2008, 4463-4465; Adv. Funct. Mater. 2010, 5, 773-780) should be cited.

Review form: Reviewer 2**Is the manuscript scientifically sound in its present form?**

Yes

Are the interpretations and conclusions justified by the results?

No

Is the language acceptable?

Yes

Is it clear how to access all supporting data?

Not Applicable

Do you have any ethical concerns with this paper?

No

Have you any concerns about statistical analyses in this paper?

No

Recommendation?

Major revision is needed (please make suggestions in comments)

Comments to the Author(s)

The submitted manuscript proposed preparation of poly(acrylic acid) coated magnetic nanospheres (PMNS) via a multiple polymerization route. In general, this work is written clearly, but some experiment design and characterization results are inadequate and the claims author proposed here don't have sufficient evidence to support. Some comments attached as follows:

- 1、 In the Introduction part, a problem was presented that magnetic nanoparticles would migrate to the surface of the nanospheres during the process of miniemulsion polymerization, and further affected the further surface modification by functional groups. Aimed to this topic, author proposed a multiple polymerization route to overcome it. However, in all subsequent experimental results described in this work, there was no direct evidence or convincing data to support the multi-step polymerization approach did not affect subsequent surface modification. For example, what is the difference in the density of the surface functional groups or the thickness of the PAA layer on the surface of the magnetic microspheres obtained from the multi-step polymerization method compared with the conventional emulsion polymer method?
- 2、 In the 3.6 Characterization, did the TGA data be measured in nitrogen atmosphere ? It should be described clearly.
- 3、 In the last sentence of 4.1.2, it should hydrodynamic size increased by 50 nm rather than size increased by 50 nm.
- 4、 In the part of 4.1.3, the saturation magnetization of MNS1 and MNS2 should directly give the actual saturation magnetization measured by VSM firstly, without subtracting the mass of ligands and polymers on the surface of magnetic particles. Only with such data could make the readers understand intuitively the magnetic properties of the obtained magnetic microspheres in this work.
- 5、 In the part of 4.2.1, MNS2 should be set as the control group when studying the particle size of microspheres varies with the increase of pH. That is to say, the particle size of MNS2 varies with the increase of pH must be measured.

Decision letter (RSOS-190141.R0)

15-Apr-2019

Dear Dr chen:

Title: Synthesis of poly(acrylic acid) coated magnetic nanospheres via a multiple polymerization route

Manuscript ID: RSOS-190141

Thank you for submitting the above manuscript to Royal Society Open Science. On behalf of the Editors and the Royal Society of Chemistry, I am pleased to inform you that your manuscript will be accepted for publication in Royal Society Open Science subject to minor revision in accordance with the referee suggestions. Please find the reviewers' comments at the end of this email.

The reviewers and handling editors have recommended publication, but also suggest some minor revisions to your manuscript. Therefore, I invite you to respond to the comments and revise your manuscript.

Please also include the following statements alongside the other end statements. As we cannot publish your manuscript without these end statements included, if you feel that a given heading

is not relevant to your paper, please nevertheless include the heading and explicitly state that it is not relevant to your work. We have included a screenshot example of the end statements for reference.

- Ethics statement

Please clarify whether you received ethical approval from a local ethics committee to carry out your study. If so please include details of this, including the name of the committee that gave consent in a Research Ethics section after your main text. Please also clarify whether you received informed consent for the participants to participate in the study and state this in your Research Ethics section.

OR

Please clarify whether you obtained the necessary licences and approvals from your institutional animal ethics committee before conducting your research. Please provide details of these licences and approvals in an Animal Ethics section after your main text.

OR

Please clarify whether you obtained the appropriate permissions and licences to conduct the fieldwork detailed in your study. Please provide details of these in your methods section.

- Data accessibility

It is a condition of publication that you make available the data and research materials supporting the results in the article. Datasets should be deposited in an appropriate publicly available repository and details of the associated accession number, link or DOI to the datasets must be included in the Data Accessibility section of the article

(<http://royalsocietypublishing.org/instructions-authors#question17>). Reference(s) to datasets should also be included in the reference list of the article with DOIs (where available).

Please include a Data Availability section after your main text stating where supporting data are available from, or where they will be made available should your article be accepted for publication.

If you wish to submit your supporting data or code to Dryad (<http://datadryad.org/>), or modify your current submission to dryad, please use the following link:
<http://datadryad.org/submit?journalID=RSOS&manu=RSOS-190141>

- Competing interests

Please include a Competing Interests section after your main text declaring any financial or non-financial competing interests. If you have no competing interests please state 'I/we have no competing interests.'

- Authors' contributions

Please include an Authors' Contributions section at the end of your main text detailing the contribution of each author. All authors should have read and approved the manuscript before submission and this should be stated in the Authors' Contributions section.

The list of Authors should meet all of the following criteria; 1) substantial contributions to conception and design, or acquisition of data, or analysis and interpretation of data; 2) drafting the article or revising it critically for important intellectual content; and 3) final approval of the version to be published.

- Acknowledgements

- Funding statement

Please include a funding section after your main text which lists the source of funding for each author.

Because the schedule for publication is very tight, it is a condition of publication that you submit the revised version of your manuscript before 24-Apr-2019. Please note that the revision deadline will expire at 00.00am on this date. If you do not think you will be able to meet this date please let me know immediately.

Supplementary files will be published alongside the paper on the journal website and posted on the online figshare repository (<https://figshare.com>). The heading and legend provided for each supplementary file during the submission process will be used to create the figshare page, so please ensure these are accurate and informative so that your files can be found in searches. Files

on figshare will be made available approximately one week before the accompanying article so that the supplementary material can be attributed a unique DOI.

Best wishes,
Dr Laura Smith
Publishing Editor, Journals

On behalf of the Subject Editor Professor Anthony Stace and the Associate Editor Professor Tobias Hertel.

Reviewer comments to Author:
Reviewer: 1

Comments to the Author(s)

In this manuscript, Chen, et al, reported a multiple polymerization route to synthesize poly (acrylic acid) coated magnetic nanospheres with controlled surface modification. The obtained magnetic nanospheres were fully characterized by XRD, TGA, TEM, DLS, and VSM. Besides, the relationship between the structure of magnetic nanospheres and the thickness of the grafted poly (acrylic acid) layer has been investigated. This work provided a new idea to control the structure and surface of magnetic polymer nanospheres. Therefore, I recommend that this work can be published in RSOS after addressing some specific issues as follows:

1. Line 13, Page 3: the description on Figure 1 is not correct. The peak positions for (422), (511), and (440) is wrong, there three peaks should be 53.7°, 57.1°, and 62.9°.
2. Page 4, line 32. Authors described the pH responsiveness of magnetic nanospheres, however, it's better to provide the hydrodynamic size in extreme pH values such as pH 3 and pH 11 in Figure 6.
3. The magnetic content of the nanospheres is low. Can the loading amounts of magnetite in polymer matrix be adjusted?
4. How the stability of the poly (acrylic acid) coated magnetic nanospheres in PBS?
5. In the caption of Figure 5, "loop" should be "loops"
6. Some related reported papers on the surface modification of magnetic composites (such as Chem. Commun. 2008, 4463-4465; Adv. Funct. Mater. 2010, 5, 773-780) should be cited.

Reviewer: 2

Comments to the Author(s)

The submitted manuscript proposed preparation of poly(acrylic acid) coated magnetic nanospheres (PMNS) via a multiple polymerization route. In general, this work is written clearly,

but some experiment design and characterization results are inadequate and the claims author proposed here don't have sufficient evidence to support. Some comments attached as follows:

- 1、 In the Introduction part, a problem was presented that magnetic nanoparticles would migrate to the surface of the nanospheres during the process of miniemulsion polymerization, and further affected the further surface modification by functional groups. Aimed to this topic, author proposed a multiple polymerization route to overcome it. However, in all subsequent experimental results described in this work, there was no direct evidence or convincing data to support the multi-step polymerization approach did not affect subsequent surface modification. For example, what is the difference in the density of the surface functional groups or the thickness of the PAA layer on the surface of the magnetic microspheres obtained from the multi-step polymerization method compared with the conventional emulsion polymer method?
- 2、 In the 3.6 Characterization, did the TGA data be measured in nitrogen atmosphere ? It should be described clearly.
- 3、 In the last sentence of 4.1.2, it should hydrodynamic size increased by 50 nm rather than size increased by 50 nm.
- 4、 In the part of 4.1.3, the saturation magnetization of MNS1 and MNS2 should directly give the actual saturation magnetization measured by VSM firstly, without subtracting the mass of ligands and polymers on the surface of magnetic particles. Only with such data could make the readers understand intuitively the magnetic properties of the obtained magnetic microspheres in this work.
- 5、 In the part of 4.2.1, MNS2 should be set as the control group when studying the particle size of microspheres varies with the increase of pH. That is to say, the particle size of MNS2 varies with the increase of pH must be measured.

Author's Response to Decision Letter for (RSOS-190141.R0)

See Appendix A.

Decision letter (RSOS-190141.R1)

28-May-2019

Dear Dr chen:

Title: Synthesis of poly(acrylic acid) coated magnetic nanospheres via a multiple polymerization route

Manuscript ID: RSOS-190141.R1

It is a pleasure to accept your manuscript in its current form for publication in Royal Society Open Science. The chemistry content of Royal Society Open Science is published in collaboration with the Royal Society of Chemistry.

On behalf of the Subject Editor Professor Anthony Stace and the Associate Editor Professor Tobias Hertel.

RSC Associate Editor
Comments to the Author:
(There are no comments.)

Reviewer(s)' Comments to Author:

Appendix A

Dear Editor,

We have revised our manuscript according to the reviewers' comments and the revised texts have been highlighted by "Track Changes" in MS Word. We appreciate the detailed and useful comments and suggestions from you and the reviewers. The point-by-point answers to the comments were listed as below.

R1:

In this manuscript, Chen, et al, reported a multiple polymerization route to synthesize poly (acrylic acid) coated magnetic nanospheres with controlled surface modification. The obtained magnetic nanospheres were fully characterized by XRD, TGA, TEM, DLS, and VSM. Besides, the relationship between the structure of magnetic nanospheres and the thickness of the grafted poly (acrylic acid) layer has been investigated. This work provided a new idea to control the structure and surface of magnetic polymer nanospheres. Therefore, I recommend that this work can be published in RSOS after addressing some specific issues as follows:

Answer: We thank this reviewer's careful reading and good understanding of our manuscript. Below we will list the corresponding revisions made in our revised manuscript to meet the reviewer's comment.

1. Line 13, Page 3: the description on Figure 1 is not correct. The peak positions for (422), (511), and (440) is wrong, there three peaks should be 53.7°, 57.1°, and 62.9°.

Answer: We thank the reviewer for pointing out this mistake. We have made corrections to Figure 1a according to the reviewer's suggestion.

2. Page 4, line 32. Authors described the pH responsiveness of magnetic nanospheres, however, it's better to provide the hydrodynamic size in extreme pH values such as pH 3 and pH 11 in Figure 6.

Answer: We thank the reviewer for his/her valuable advice. In Figure 6, we added hydrodynamic sizes at extreme pH (pH 3 and pH 11). In line 9, section 4.2.1, the discussion for size at extreme pH is added as "PMNS is unstable at low pH ($\text{pH} \leq 4$) because of insufficient space hindrance and reduced electrostatic repulsion."

3. The magnetic content of the nanospheres is low. Can the loading amounts of magnetite in polymer matrix be adjusted?

Answer: Actually, the amount of magnetic content can be adjusted. However, we take the magnetic content of 5wt% as an example to study the effect of multi-step polymerization on the functional modification of magnetic microspheres. In our future work, we will focus on the magnetic content effect on the preparation process.

4. How the stability of the poly (acrylic acid) coated magnetic nanospheres in PBS?

Answer: Poly (acrylic acid) coated magnetic nanospheres can be stored for up to 6 months in PBS solution without precipitation. We added "The MNS2 showed a large amount of precipitation in one month. While the PMNS stored in a PBS buffer was stable for up to 6 months without precipitation, which demonstrated that the poly(acrylic acid) chains endow PMNS a good stability." in line 10, section 4.2.1 in the revised manuscript.

5. In the caption of Figure 5, "loop" should be "loops"

Answer: We thank the reviewer for pointing out the typo and it has been corrected.

6. Some related reported papers on the surface modification of magnetic composites (such as Chem. Commun. 2008, 4463-4465; Adv. Funct. Mater. 2010, 5, 773-780) should be cited.

Answer: We thank the reviewers for their valuable suggestions for related references. We have carefully reviewed the literatures and cited them in the revised manuscript as **ref [15]** in line 12, section 2, and **ref [17]** in line 12, section 2, respectively.

R2:

The submitted manuscript proposed preparation of poly(acrylic acid) coated magnetic nanospheres (PMNS) via a multiple polymerization route. In general, this work is written clearly, but some experiment design and characterization results are inadequate and the claims author proposed here don't have sufficient evidence to support. Some comments attached as follows:

Answer: We thank this reviewer's careful reading and good understanding of our manuscript. Below we will list the corresponding revisions made in our revised manuscript to meet the reviewer's comment.

1. In the Introduction part, a problem was presented that magnetic nanoparticles would migrate to the surface of the nanospheres during the process of miniemulsion polymerization, and further affected the further surface modification by functional groups. Aimed to this topic,

author proposed a multiple polymerization route to overcome it. However, in all subsequent experimental results described in this work, there was no direct evidence or convincing data to support the multi-step polymerization approach did not affect subsequent surface modification. For example, what is the difference in the density of the surface functional groups or the thickness of the PAA layer on the surface of the magnetic microspheres obtained from the multi-step polymerization method compared with the conventional emulsion polymer method?

Answer: We thank the reviewer for this question. Here, we are sorry for the reviewer's confusion on the motivation of our manuscript because of our improper presentation. In this work, poly(acrylic acid) coated magnetic nanospheres were prepared by a multi-polymerization method, which is an alternative route for preparation of functional magnetic nanospheres with controlled structure. It is also proposed that the seed emulsion polymerization method can be used to control the aggregation morphology of MNP in magnetic microspheres and the distribution of MNP to affect the surface functionalization. To avoid the confusion, we revised some sentences in the introduction. In line 22, section 2, We have deleted the " **However, MNP would migrate to the surface of the nanospheres in some cases, which affect the further surface modification by functional groups [37]. To avoid this negative effect, efforts have been made by researchers.**" Then, in line 24, section 2, add "**In order to further improve the encapsulation of magnetic nanoparticles**" later. In line 28, section 2, add "**Therefore, we provide an alternative way to prepare functional magnetic nanospheres.**" In line 33, section 2, add "**At the same time, the effect of aggregation state of magnetic nanoparticles in magnetic nanospheres on surface functionalization was also studied.**" In order to make the paper's data more convincing, we compare it with traditional aggregation methods. In line 15, section 4.2.2, add "**Xu used a seed emulsion polymerization method to increase the thickness of functional layer by ~10 nm [19]. Compared with the literature, our method improves the layer thickness to some extent.**"

2. In the 3.6 Characterization, did the TGA data be measured in nitrogen atmosphere ? It should be described clearly.

Answer: TGA was carried out in air atmosphere instead of nitrogen atmosphere. In line 3, section 3.6, "The thermogravimetric analysis (TGA) model Q600 SD was used to analyze the magnetite content at a heating rate of 10 °C/min." was corrected to "**The thermogravimetric analysis (TGA) model Q600 SD was used to analyze the magnetite content at a heating rate of 10 °C/min in air atmosphere.**"

3. In the last sentence of 4.1.2, it should hydrodynamic size increased by 50 nm rather than size increased by 50 nm.

Answer: It has been corrected according to the reviewer's suggestion in the revised manuscript.

4. In the part of 4.1.3, the saturation magnetization of MNS1 and MNS2 should directly give the actual saturation magnetization measured by VSM firstly, without subtracting the mass of ligands and polymers on the surface of magnetic particles. Only with such data could make the readers understand intuitively the magnetic properties of the obtained magnetic microspheres in this work.

Answer: We thank the reviewer for pointing out this question. We provided a more intuitive data description as shown in Figure 5a. Add " **In Figure 5a, the saturation magnetizations of MNP, MNS1 and MNS2 are 52.4 emu/g, 2.7 emu/g and 0.96 emu/g, respectively.** " in line 2, section 4.1.3.

5. In the part of 4.2.1, MNS2 should be set as the control group when studying the particle size of microspheres varies with the increase of pH. That is to say, the particle size of MNS2 varies with the increase of pH must be measured.

Answer: We did the control experiments and added the data to Figure 6b. In line 7, section 4.2.1, we added text as "**With the increase of pH value, the hydrodynamic size of MNS2 is almost a constant, because MNS2 has no pH sensitivity.**"